# Validation of Novel Molecular Imaging Targets Identified by Functional Genomic mRNA Profiling to Detect Dysplasia in Barrett’s Esophagus

**DOI:** 10.3390/cancers14102462

**Published:** 2022-05-17

**Authors:** Xiaojuan Zhao, Ruben Y. Gabriëls, Wouter T. R. Hooghiemstra, Marjory Koller, Gert Jan Meersma, Manon Buist-Homan, Lydia Visser, Dominic J. Robinson, Anna Tenditnaya, Dimitris Gorpas, Vasilis Ntziachristos, Arend Karrenbeld, Gursah Kats-Ugurlu, Rudolf S. N. Fehrmann, Wouter B. Nagengast

**Affiliations:** 1Department of Gastroenterology and Hepatology, University Medical Center Groningen, University of Groningen, 9713 GZ Groningen, The Netherlands; x.zhao01@umcg.nl (X.Z.); r.y.gabriels@umcg.nl (R.Y.G.); w.t.r.hooghiemstra@umcg.nl (W.T.R.H.); g.j.meersma@umcg.nl (G.J.M.); m.buist-homan@umcg.nl (M.B.-H.); 2Cancer Research Center Groningen, Department of Medical Oncology, University Medical Center Groningen, University of Groningen, 9713 GZ Groningen, The Netherlands; r.s.n.fehrmann@umcg.nl; 3Department of Clinical Pharmacy and Pharmacology, University Medical Center Groningen, University of Groningen, 9713 GZ Groningen, The Netherlands; 4Department of Surgery, University Medical Center Groningen, University of Groningen, 9713 GZ Groningen, The Netherlands; marjorykoller@gmail.com; 5Department of Laboratory Medicine, University Medical Center Groningen, University of Groningen, 9713 GZ Groningen, The Netherlands; 6Department of Pathology and Medical Biology, University Medical Center Groningen, University of Groningen, 9713 GZ Groningen, The Netherlands; l.visser@umcg.nl (L.V.); a.karrenbeld@umcg.nl (A.K.); g.kats-ugurlu@umcg.nl (G.K.-U.); 7Center for Optic Diagnostics and Therapy, Erasmus University Medical Center, 3015 GD Rotterdam, The Netherlands; d.robinson@erasmusmc.nl; 8Chair of Biological Imaging, Central Institute for Translational Cancer Research (TranslaTUM), School of Medicine, Technical University of Munich, 80333 Munich, Germany; anna.tenditnaya@helmholtz-muenchen.de (A.T.); dimitrios.gkorpas@helmholtz-muenchen.de (D.G.); v.ntziachristos@tum.de (V.N.); 9Institute of Biological and Medical Imaging, Helmholtz Zentrum München (GmbH), 85764 Neuherberg, Germany

**Keywords:** Barrett’s esophagus, esophageal adenocarcinoma, fluorescent molecular endoscopy, novel biomarkers, functional genomic mRNA profiling

## Abstract

**Simple Summary:**

Barrett’s esophagus (BE) is the precursor of esophageal adenocarcinoma (EAC). Dysplastic BE (DBE), including low-grade dysplasia (LGD) and high-grade dysplasia (HGD), shows a higher progression risk to EAC compared to non-dysplastic BE (NDBE). If LGD or HGD is detected, more intensive endoscopic surveillance or endoscopic treatment is recommended. This results in a significantly improved prognosis compared to EACs treated by surgery and/or chemoradiotherapy. However, the miss rates for detecting DBE by endoscopy remain high. Fluorescence molecular endoscopy (FME) can fill this gap by targeting the tumor-specific expression of proteins. This study aimed to identify target proteins suitable for FME. We identified SPARC, SULF1, PKCι, and DDR1 as promising imaging targets for FME to differentiate DBE from NDBE tissue. We are also the first to develop near-infrared fluorescent tracers, SULF1-800CW and SPARC-800CW, for the endoscopic imaging of DBE tissue.

**Abstract:**

Barrett’s esophagus (BE) is the precursor of esophageal adenocarcinoma (EAC). Dysplastic BE (DBE) has a higher progression risk to EAC compared to non-dysplastic BE (NDBE). However, the miss rates for the endoscopic detection of DBE remain high. Fluorescence molecular endoscopy (FME) can detect DBE and mucosal EAC by highlighting the tumor-specific expression of proteins. This study aimed to identify target proteins suitable for FME. Publicly available RNA expression profiles of EAC and NDBE were corrected by functional genomic mRNA (FGmRNA) profiling. Following a class comparison between FGmRNA profiles of EAC and NDBE, predicted, significantly upregulated genes in EAC were prioritized by a literature search. Protein expression of prioritized genes was validated by immunohistochemistry (IHC) on DBE and NDBE tissues. Near-infrared fluorescent tracers targeting the proteins were developed and evaluated ex vivo on fresh human specimens. In total, 1976 overexpressed genes were identified in EAC (*n* = 64) compared to NDBE (*n* = 66) at RNA level. Prioritization and IHC validation revealed SPARC, SULF1, PKCι, and DDR1 (all *p* < 0.0001) as the most attractive imaging protein targets for DBE detection. Newly developed tracers SULF1-800CW and SPARC-800CW both showed higher fluorescence intensity in DBE tissue compared to paired non-dysplastic tissue. This study identified SPARC, SULF1, PKCι, and DDR1 as promising targets for FME to differentiate DBE from NDBE tissue, for which SULF1-800CW and SPARC-800CW were successfully ex vivo evaluated. Clinical studies should further validate these findings.

## 1. Introduction

In 2018, the Netherlands had the highest incidence rates of esophageal adenocarcinoma (EAC) worldwide, with an age-standardized incidence rate (ASR) of 4.4 per 100,000 [1]. Barrett’s esophagus (BE) is a change in the lining of the distal esophagus that stratifies the squamous epithelium, replaced by a metaplastic columnar epithelium. BE is the precursor stage of EAC. The usual progression is stepwise from non-dysplastic BE (NDBE) to low-grade dysplasia (LGD), to high-grade dysplasia (HGD), and finally to EAC. LGD shows an annual progression rate to HGD or EAC of approximately 11.8%, compared to 0.33% for NDBE [2,3,4]. The clinical guidelines of the American Gastroenterological Association (AGA) recommend repeated endoscopic surveillance for LGD and endoscopic treatment for HGD (p Tis N0 M0) and mucosal EAC (p T1a N0 M0) [5,6,7]. Once the tumor cells invade the submucosal or deeper layers, the American Society of Clinical Oncology (ASCO) recommends esophagectomy and/or chemoradiotherapy [8]. Although the five-year overall survival rate of esophageal adenocarcinoma is around 20% [9], if tumor cells are limited to the mucosal layer (p Tis-1a N0 M0) or superficial submucosal layer (p T1b sm1 N0 M0), this rate can be up to 93% after endoscopic treatment [10]. Therefore, early detection and subsequent endoscopic surveillance or treatment of dysplastic BE (DBE, including LGD and HGD) would improve the prognosis and avoid the burden of extensive surgery and/or chemoradiotherapy [10]. Hence, the ability to accurately discriminate DBE from NDBE is of vital importance.

According to the Seattle protocol, the current gold standard for detecting DBE is 4-quadrant biopsies every 1–2 cm Barrett esophagus under high-definition white light (HDWL) endoscopy [11]. Due to sampling error, poor adherence to the protocol, or interobserver variation, the sensitivity of the Seattle protocol to detect DBE is approximately 64% to 73% [12,13]. Consequently, 24% of EACs are diagnosed within one year of a prior diagnosis of NDBE by endoscopic surveillance. This indicates a high miss rate of DBE and perhaps even EACs during surveillance endoscopy [14]. The high miss rate under HDWL endoscopy underlines the importance of improving endoscopic methods for detecting DBE and mucosal EACs.

The progression from NDBE to DBE and EAC is mediated by genomic alterations, such as the loss of heterozygosity of tumor suppressor genes TP53 and CDKN2A, increased copy numbers of oncogenes VEGFA and PRKCI, and upregulation of the E-cadherin signaling pathway [15,16]. Genomic variations that frequently occur in DBE and EAC can sometimes lead to downstream overexpression at the protein level. Fluorescence molecular endoscopy (FME) can target the overexpressed proteins by fluorescence tracers, thus highlighting the presence of dysplasia and carcinoma lesions. Our feasibility phase 1 study has shown that even flat and difficult-to-distinguish DBE lesions and mucosal EACs are identifiable with FME. The topical administration of bevazucimab-800CW targeting vascular endothelial growth factor A (VEGF-A) was able to improve early lesion detection by ~33% compared with HD-WL endoscopy and narrowband imaging [17]. Due to the intra- and intertumor heterogeneity of DBE and EAC lesions, it is of great importance to identify novel imaging protein targets specifically overexpressed in DBE and EAC lesions [18]. The development of a multispectral FME system technically would enable the targeting of multiple proteins concurrently [19].

In this feasibility study, we therefore aimed to identify novel imaging protein targets which have the potential to improve the detection of DBE and mucosal EAC lesions by FME. At first, we collected microarray expression profiles of patient-derived EAC samples and NDBE samples from the Gene Expression Omnibus database. Secondly, a bioinformatic algorithm named functional genomic mRNA (FGmRNA) profiling was applied to correct gene expression profiles for non-genetic factors and to predict target upregulation on a protein level [20]. Thirdly, the protein products of predicted upregulated genes were prioritized based on a literature search and immunohistochemistry validation to select promising protein targets for imaging. Near-infrared fluorescence tracers targeting the proteins were developed and validated ex vivo.

## 2. Materials and Methods

### 2.1. Identification of Differentially Expressed Genes with Functional Genomic mRNA Profiling

Publicly available microarray expression profiles of patient-derived primary EAC and NDBE samples generated with the Affymetrix HG-U133 plus 2.0 and the HG-U133A platforms were obtained from the Gene Expression Omnibus database [21]. Pre-processing, normalization, and quality control were performed as previously described [22]. Subsequently, FGmRNA profiling was applied to filter out non-genetic factors and extract the downstream consequences of genetic alterations on gene expression profiles [20]. FGmRNA profiling is a bioinformatic algorithm for correcting gene expression profiles, thus allowing for an enhanced view on the downstream effects of genomic alterations on gene expression levels. To identify upregulated genes in EAC, a transcriptome-wide class comparison (Welch’s *t*-test) between FGmRNA profiles of EAC and NDBE was performed. A multivariate permutation (MVP) test was performed to control the false discovery rate (false discovery rate 5%; confidence level 80%, 1000 permutations).

### 2.2. Prioritization Strategy

Based on the FGmRNA profiling results, the top 150 predicted upregulated genes in EAC compared to NDBE were selected for further prioritization through a database search and a literature search. At first, the protein products of genes should be expressed on the plasma membrane to be easily reached by fluorescence tracers. The probable subcellular localization of the protein products of genes was identified in three databases, including the Human Protein Atlas, GeneCards, and GenetICA-Network [23]. Only the genes whose protein products are probably localized on the plasma membrane were selected for a literature search. Regarding the literature search, we searched per gene in PubMed for articles published in English from conception until December 2020. Secondly, we searched PubMed for the function of the gene and its relationship to carcinogenesis. The genes with a biological pathway involved with carcinogenesis were prioritized for immunohistochemical validation. Thirdly, we searched PubMed to identify genes with known upregulated protein expression in gastrointestinal cancer using the following search terms: HUGO gene symbol of the target under investigation in combination with ‘immunohistochemistry’ and ‘esophageal cancer’ or ‘gastrointestinal cancer’ or ‘gastric cancer’ or ‘colorectal cancer’ or ‘rectum cancer’. Finally, we searched if there was published molecular imaging research targeting the protein products of the genes. Any gene with its protein product targeted by a molecular imaging tracer and already studied in esophageal adenocarcinoma or Barrett’s esophagus was discarded. 

### 2.3. Patient Inclusion and Specimen Selection

For the pre-analysis of this study to analyze protein expression using IHC, we included 25 patients (19 males and 6 females, mean age 67.88 years old, all Caucasian), diagnosed with a dysplastic esophageal lesion and scheduled for an endoscopic resection. In total, 67 endoscopic mucosal resection (EMR) specimens from 25 patients were collected and embedded into FFPE coupes. For the details of patient numbers, demographic characteristics, and histological diagnosis, please see Appendix A. 

For the analysis, we selected coupes containing both dysplastic tissue and non-dysplastic tissue to evaluate protein expression; in total, we selected 73 coupes. H&E and anti-P53 staining was performed to distinguish non-dysplastic tissue from dysplastic tissue. All specimens were analyzed by our BE expert pathologists, GK-U and AK, independently. We selected 264 histopathological areas, including normal squamous epithelium tissue (*n* = 44), pre-existing normal glands (*n* = 8), normal stomach tissue (*n* = 15), NDBE tissue (*n* = 59), LGD tissue (*n* = 37), HGD tissue (*n* = 55), and EAC tissue (*n* = 46).

### 2.4. Immunohistochemistry

Immunohistochemistry (IHC) was carried out on formalin-fixed and paraffin-embedded (FFPE) tissue blocks derived from the clinical ESCEND study (ClinicalTrials.gov: NCT03877601), performed at the University Medical Center Groningen (UMCG). All human tissue samples were handled in accordance with the guidelines of the UMCG ethics board. The staining protocols were optimized with the help of gastrointestinal pathologist AK and IHC expert LV (Appendix A). Negative controls using 1% bovine serum albumin (BSA) in phosphate-buffered saline (PBS) were performed. The IHC slices were scanned with Hamamatsu NanoZoomer and viewed with NDP.view 2 (Hamamatsu Photonics, Japan). IHC slices were manually quantified with the H-score method by XZ and RYG independently [24]. The equation to calculate H-score is H-Score = 1 * (percentage of cells weakly stained) + 2 * (percentage of cells moderately scored) + 3 * (percentage of cells strongly stained). 

### 2.5. Tracer Construction

Tracers were constructed by conjugating a near-infrared (NIR) fluorescent dye to an antibody via an NHS–ester binding. We used a polyclonal antibody targeting SULF1 (Invitrogen, Waltham, MA, USA) and a monoclonal antibody targeting SPARC (Invitrogen, Waltham, MA, USA). For both antibodies, we used 500 µg in a concentration of 1 mg/mL and 2 mg/mL for SULF1 and SPARC, respectively. As a fluorescent dye, we used IRDye 800CW NHS ester (LI-COR Biosciences, Lincoln, NE, USA).

We first purified and buffer-exchanged the antibodies to a sodium phosphate buffer 50 mM pH 8.5 using a PD MiniTrap column with Sephadex G-25 resin for the conjugation. After the buffer exchange, dye was added in a molar dye:antibody ratio of 2:1 and left to conjugate for at least 1 h at room temperature, protected from light. After the conjugation, the tracers were purified of excess dye and buffer exchanged with a sodium phosphate buffer 50 mM pH 7.0 for storage. Tracers were diluted with the same sodium phosphate buffer pH 7.0 to a 0.1 mg/mL concentration.

### 2.6. Ex Vivo Fluorescence Imaging and Validation of Fluorescence Intensities

To test the applicability of SULF1-800CW and SPARC-800CW, ex vivo fluorescence imaging was performed on EMR specimens. Fresh EMR specimens were collected immediately after resection. All patients received oral and written information before giving consent. After multi-diameter single-fiber reflectance/single-fiber fluorescence (MDSFR/SFF) spectroscopy measurements, the EMR specimens were incubated with 1 mL of 0.1 mg/mL of SULF1-800CW or SPARC-800CW for 5 min. Fluorescence images were subsequently acquired of EMR specimens with the PEARL Trilogy Imaging System (LI-COR Biosciences, Lincoln, NE, USA). After rinsing with saline three times, white-light images and fluorescence images were acquired of EMR specimens with the PEARL Trilogy Imaging System again. The MDSFR/SFF spectroscopy measurements were performed again. EMR specimens were fixated in 10% neutral buffered formalin (NPF) for 24 h. Then, EMR specimens were inked blue on the three o’clock side, black on the nine o’clock side, and subsequently cut into bread-loaf slices (BLS) with a width of around 0.5 cm. Subsequently, the BLS were imaged on both the mucosal and luminal sides using an Odyssey CLx flatbed scanner (LI-COR Biosciences, Lincoln, NE, USA). Finally, these BLS were sent back to the Pathology Department of the UMCG and diagnosed by board-certified pathologists blinded to the fluorescent images. The BLS were cut into 4 µm sections, on which hematoxylin and eosin (H&E) staining was performed. Anti-P53 IHC staining was performed when pathologists could not diagnose whether the lesion was dysplastic based on cytological and morphological changes. Anti-caldesmon IHC staining was performed to investigate if there was tumor cell infiltration in the muscularis mucosae or muscularis propria. FFPE blocks and slides were imaged to determine fluorescent intensities with the Odyssey CLx flatbed scanner. To calculate the mean fluorescence intensity (MFI) of DBE on one EMR specimen, the DBE area was delineated on the EMR specimen according to the histology diagnosis by pathologists AK and GK-U. Pathologists AK and GK-U diagnosed NDBE, LGD, HGD, and EAC according to the Vienna Classification [25]. As the coagulation effect happens during EMR procedures, the edge of each EMR specimen was cut off according to the location of the pins and was not calculated by MFI. Appendix A shows how the histology-correlated MFI was calculated on each EMR specimen.

### 2.7. MDSFR/SFF Spectroscopy

EMR specimens were subjected to MDSFR/SFF spectroscopy before and after tracer incubation, and after rinsing with phosphate-buffered saline. MDSFR measured the tissue absorption coefficient (μ_a_) and the reduced scattering coefficient (μ_s_’). The fluorescence values measured by SFF spectroscopy were corrected by μ_a_ and μ_s_’ to obtain the total intrinsic fluorescence (Qμ^f^_a,x_) of the fluorophores of SULF1-800CW or SPARC-800CW in situ. All measurements were repeated in triplicate to obtain a median Qμ^f^_a,x_ value per location [26].

### 2.8. Statistical Analyses

A two-tailed non-parametric Mann–Whitney U test (IBM SPSS 22.0.0; IBM Corporation, Armonk, NY, USA) was used to compare mean H-scores. Mean H-scores were based on the independent H-scores provided by the two researchers, XJZ and RYG. Spearman correlation between the paired H-scores obtained from the two independent researchers was used to assess the inter-observer agreement in manual H scoring. *p* < 0.05 was considered statistically significant for the H-score analysis.

## 3. Results

### 3.1. Class Comparison Identifies Upregulated Genes in Esophageal Adenocarcinoma

FGmRNA profiling was applied to 66 NDBE samples and 64 EAC lesions [27,28,29,30,31,32]. We identified 1976 genes that showed FGmRNA overexpression in EAC samples compared to NDBE tissue (FDR 5%, CL 80%, Appendix A). Well-known tumor-related genes were found in this ranking, such as ERBB3 (24th rank), VEGFA (75th rank), and ERBB2 (365th rank). The first 500 ranked genes are shown in Appendix A.

### 3.2. Target Selection in Dysplastic BE: Nine Possible Imaging Targets Identified

The prioritization strategy was applied to the top 150 upregulated genes (Appendix A) [33,34,35,36,37,38,39,40,41,42,43,44,45,46,47,48,49,50,51,52,53,54,55,56,57,58,59,60,61,62,63,64,65,66,67,68,69,70,71,72,73,74,75,76,77,78]. After searching in the three databases, we identified 37 genes whose encoding proteins were predicted to localize on the plasma membrane. These 37 genes were searched in PubMed, and 21 genes were found to be related to carcinogenesis in general. Moreover, 15 genes have known downstream overexpression in gastrointestinal carcinoma at the protein level. These 15 genes could be of interest for DBE and mucosal EAC. The protein products of eight genes have shown overexpression in neoplasia of the esophagus. The encoding proteins of five genes had been tested previously as molecular imaging targets. 

For further validation by immunohistochemistry, we selected proteins related to carcinogenesis, predicted as membranous localized, with known overexpression in esophageal cancer or gastrointestinal cancer, and without previous optical molecular imaging research in EAC or Barrett’s esophagus. This filtering resulted in protein products of nine genes selected out of the top 150 genes, including GREM1, SPARC, ENPEP, SULF1, ERBB3, PRKCI, CDH11, DDR1, and TMPRSS4.

### 3.3. The Potential Imaging Targets SULF1, SPARC, DDR1, and PKCι

We analyzed protein expression with immunohistochemistry (IHC) for the nine candidate genes to determine protein expression differences between non-dysplastic and dysplastic human BE tissue. The IHC protocols are provided in Appendix A. IHC was first performed in a small cohort of human dysplastic (*n* = 5) and non-dysplastic BE tissue (*n* = 5). Five out of nine proteins showed clear protein overexpression in human dysplastic BE tissue compared to non-dysplastic BE tissue. These proteins are extracellular sulfatase Sulf-1 (SULF1), secreted protein acidic and cysteine rich (SPARC), epithelial discoidin domain-containing receptor 1 (DDR1), protein kinase C iota type (PKCι), and transmembrane protease serine 4 (TMPRSS4). These proteins were subsequently analyzed by IHC in a larger cohort of 25 BE patients diagnosed with a dysplastic lesion. All five proteins showed plasma membranous and cytoplasmic staining in human dysplastic BE tissue (Appendix A).

Representative images of both non-dysplastic BE and dysplastic BE are shown in Figure 1a–d. The delineated DBE areas showed intermediate to high expression of the proteins SULF1, DDR1, PKCι, and SPARC, while NDBE areas showed low or negative expression (Figure 1). SULF1 (Figure 1a), DDR1 (Figure 1b), PKCι (Figure 1c), and SPARC (Figure 1d) showed differential protein expression between DBE and NDBE. H-score quantification showed that most dysplastic BE samples had intermediate and high positive epithelial staining (SULF1, 95.7%; PKCι, 62.0%; DDR1, 99.3%; SPARC, 94.9%; Figure 2, Table 1). The adjacent non-dysplastic tissue in BE samples showed low to negative SULF1 expression (93.2% of samples) and PKCι expression (85.7% of samples) (Figure 2, Table 2). Intermediate to low expression of DDR1 (91.5% of samples; Figure 2, Table 2) and SPARC (93.1% of samples; Figure 2, Table 2) was observed in adjacent non-dysplastic tissue in BE samples. Of these four targets, SULF1 showed the largest difference in H-score between DBE and NDBE tissue. Furthermore, all four targets showed differential expression not only in non-dysplastic BE and high-grade dysplasia and adenocarcinoma (*p* < 0.001, HGD/EAC to NDBE; Figure 1a–d), but also in low-grade dysplasia within Barrett’s esophagus (*p* < 0.001, LGD to NDBE; Figure 1a–d).

Stronger TMPRSS4 expression was observed in delineated non-dysplastic tissue compared to dysplastic tissue (Appendix A). H-score quantification showed an intermediate to high anti-TMPRSS4 staining in the majority of both dysplastic (99.3%) and non-dysplastic (100%) BE tissue samples (Appendix A). TMPRSS4 showed no significant difference in mean H-score between DBE and NDBE tissue samples (*p* = 0.60).

The H-scores generated by the two independent researchers showed Spearman R > 0.95 for all five antibodies of proteins SULF1, DDR1, PKCι, TMPRSS4, and SPARC (Appendix A), indicating the scoring consistency between the two independent researchers (Figure 1).

### 3.4. SPARC-800CW and SULF1-800CW Tracers Can Be Used for Fluorescence Imaging of Malignant and Premalignant EAC Lesions

After tissue incubation with SPARC-800CW or SULF1-800CW and rinsing, dysplastic BE tissue was highly fluorescent, while non-dysplastic tissue showed minimal fluorescence (Figure 3). The fluorescence images and delineations of dysplastic BE areas on each fresh EMR specimen (4 EMRs from 2 patients, both males, mean age 63 years old, both Caucasian, for SULF1-800CW; 4 EMRs from 3 patients, 2 males and 1 female, mean age 67 years old, all Caucasian, for SPARC-800CW) are presented in Appendix A. Due to the coagulation effect, the tissue on the edges of EMR specimens was more accessible for the tracers to penetrate, which resulted in high fluorescence on the edges of these specimens (Appendix A).

The mean fluorescence intensity (MFI) was calculated in tissue without coagulation. Higher MFI of SPARC-800CW was seen in dysplastic BE tissue (median 0.2185 arbitrary units (a.u.)) compared with non-dysplastic tissue (median 0.1650 a.u.). After tissue incubation with SULF1-800CW and rinsing, the median MFI of SULF1-800CW in DBE tissue (median 0.04250 a.u.) was higher than in non-dysplastic tissue (median 0.03050 a.u.). The paired data analysis showed that the MFI in dysplastic BE tissue was always higher compared to the MFI in non-dysplastic tissue from the same EMR specimens (Figure 4c).

After tissue incubation and rinsing, the intrinsic fluorescence of each EMR specimen quantified by MDSFR/SFF spectroscopy, correcting for optical properties such as absorption and scattering, was higher than before incubation. Fluorescence also decreased after rinsing (Figure 4a). The MFI in each EMR specimen (Figure 4b) showed the same trend as the intrinsic fluorescence measurements.

## 4. Discussion

Early detection and treatment of dysplastic Barrett’s esophagus could prevent its progression into esophageal adenocarcinoma. Fluorescence molecular endoscopy has the potential to improve the detection of dysplastic BE and mucosal EAC by targeting tumor-specific biomarkers at protein level. This study presents an extensive list of FGmRNA-overexpressing genes. The protein products of these genes can serve as promising imaging targets for detecting dysplastic lesions in the esophagus. Our immunohistochemistry results demonstrate the significant protein overexpression of SULF-1, SPARC, PKCι, and DDR1 in dysplastic lesions compared to NDBE tissue. More importantly, we succeeded in developing fluorescently labeled markers to target SULF-1 and SPARC and showed their increased uptake in Barrett’s dysplasia. This demonstrates the potential for using this approach with FME in Barrett surveillance to improve dysplasia detection. 

To the best of our knowledge, we are the first to perform FGmRNA profiling to identify novel markers that could be used for tracer development in combination with NIR-FME to improve early EAC lesion detection in patients with a Barrett’s esophagus. *FZD5*, *WNT2*, and *LRP5* are among the most highly upregulated genes in FGmRNA profiles and are known to be involved in the Wnt signaling pathway. The abnormal activation of this pathway plays a vital role in the oncogenesis of colorectal cancer [79], and it has also been shown to be involved in the progression from Barrett’s metaplasia to dysplasia [80]. The four protein targets we identified here—SULF1, SPARC, PKCι, and DDR1—all participate in carcinogenesis. SULF1 belongs to the endosulfatase family and can hydrolyze the sulfate residues of heparan sulfate proteoglycans (HSPGs) to modulate the role of HSPGs in carcinogenesis [81]. As a multifunctional matricellular glycoprotein, SPARC plays a role in tumorigenesis and metastasis through regulating the cell–matrix interaction [82]. PKCι is a serine/threonine protein kinase and its encoding gene, *PRKCI*, is a well-known oncogene. PKCι maintains *PRKCI* copy number gains in cancer cells and contributes to tumor initiation [83]. The receptor tyrosine kinase DDR1 has been shown to be vitally important in the proliferation, migration, and metastasis of cancer cells [84]. Therefore, FGmRNA profiling not only identified the possible imaging targets for dysplastic BE, but also reflected the underlying biology related to carcinogenesis.

In the current study, we demonstrated the overexpression at the protein level of SULF1, PKCι, and DDR1 in dysplastic tissue compared to non-dysplastic BE tissue. Chun-Tao et al. showed the overexpression of SULF1 in esophageal squamous cell carcinoma (ESCC) by immunohistochemistry, and their localization of the protein in the membrane is in accordance with our results [85]. The overexpression of DDR by immunohistochemistry was also reported in ESCC, whereby the protein was localized in the cytoplasm [62]. However, there are two subtypes of DDR protein—DDR1 and DDR2—and the subtype was not described in the latter study. We found that DDR1 showed both membranous and cytoplasmic localization (Appendix A). 

With the aim of distinguishing dysplastic BE from non-dysplastic BE tissue, we identified and validated three novel protein markers: SULF1, DDR1, and PKCι. Natalia et al. showed that SPARC is overexpressed at the mRNA level in esophageal adenocarcinoma compared to Barrett’s esophagus [86]. John et al. first showed the overexpression of SPARC at the protein level in esophageal adenocarcinoma, which is consistent with our results [37]. More importantly, the development and ex vivo use of SULF1-800CW and SPARC-800CW further support the potential use of the two tracers as imaging probes for fluorescence molecular endoscopy. A multiplexed FME system enabling the simultaneous targeting of two molecules was successfully developed by Wang’s group [19]. Recently, we also developed a multispectral FME system in cooperation with the Technical University of Munich (Germany). The concurrent imaging of SULF1-800CW and SPARC-800CW has potential to further improve the wide-field endoscopic diagnosis of dysplastic BE.

The ex vivo analysis with our novel tracers was performed in a small sample size of freshly selected esophageal specimens (*n* = 4 for SULF1-800CW from 2 patients and *n* = 4 for SPARC-800CW from 3 patients). In this feasibility study, we tested these newly developed tracers in a small sample size, with the inherent limitations. However, validating these results in a larger sample size in the near future is warranted. Another consideration for future research concerns the administration method. Compared to in vivo administration or in vivo spraying of the tracers, ex vivo spraying may weaken the distribution and binding of the tracers to target proteins due to the loss of cell activity [87]. Better performance of cGMP-developed SULF1-800CW and SPARC-800CW to detect dysplastic BE could be expected in an in vivo study. In a later stage, human studies should be conducted to further validate the potential of SULF1-800CW and SPARC-800CW as imaging tracers for fluorescence molecular endoscopy.

## 5. Conclusions

In the current study, we identified new imaging targets which could be used for the early detection of premalignant esophageal lesions in BE by FGmRNA profiling. We validated four new imaging markers—SULF1, SPARC, PKCι, and DDR1—that significantly distinguish dysplastic from non-dysplastic BE tissue. We also demonstrated the feasibility of using these markers for FME by the development and application of two novel fluorescently labeled tracers: SULF1-800CW and SPARC-800CW. Concurrent imaging of SULF1-800CW and SPARC-800CW in a large cohort is of interest for further research.

## Figures and Tables

**Figure 1 cancers-14-02462-f001:**
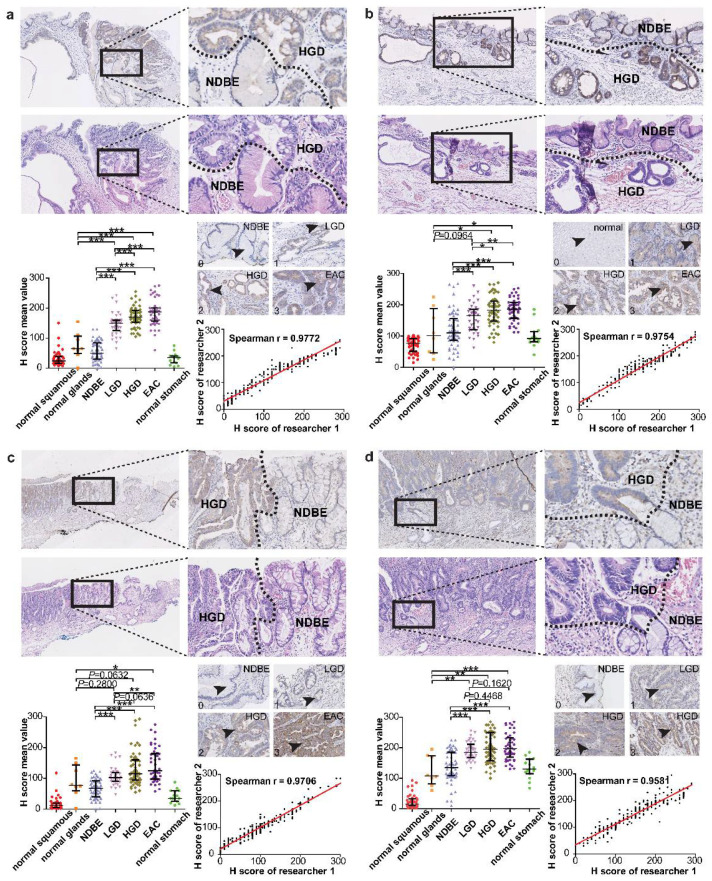
Immunohistochemistry results of SULF1, DDR1, PKCι, and SPARC. Representative images and statistics of SULF1 (**a**), DDR1 (**b**), PKCι (**c**), SPARC (**d**) are shown. In each section, the upper panel depicts the IHC images in low and high magnification, accompanied by H&E staining results in serial cutting slices and pathological delineation. The left lower panel shows the comparison of the mean H-score in seven groups: normal squamous epithelium, normal glands, NDBE, LGD, HGD, EAC, and normal stomach. Two researchers performed H-scoring separately and blindly. The mean H-score is the average value of the H-scores of researcher I and researcher II. The middle right panel explains how we defined the staining intensity of 0 to 3. No IHC positive staining is scored as 0. Weak IHC positive staining is scored as 1. Moderate IHC positive staining is scored as 2. Strong IHC positive staining is scored as 3. The lower right panel shows the correlation analysis of H-scoring by researcher I and researcher II, with Spearman correlation. EAC: esophageal adenocarcinoma; HGD: high-grade dysplasia; LGD: low-grade dysplasia; NDBE: non-dysplastic Barrett’s esophagus. * *p* value < 0.01; ** *p* value < 0.001; *** *p* value < 0.0001.

**Figure 2 cancers-14-02462-f002:**
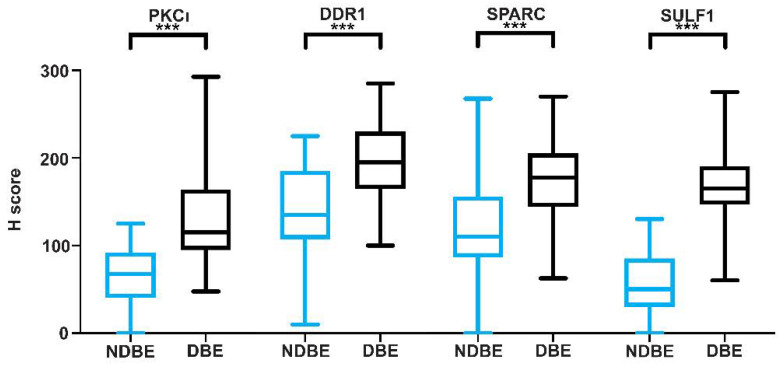
The comparison of immunohistochemistry results between dysplastic Barrett’s esophagus and non-dysplastic Barrett’s esophagus. DBE: dysplastic Barrett’s esophagus; NDBE: non-dysplastic Barrett’s esophagus. *** *p* value < 0.0001.

**Figure 3 cancers-14-02462-f003:**
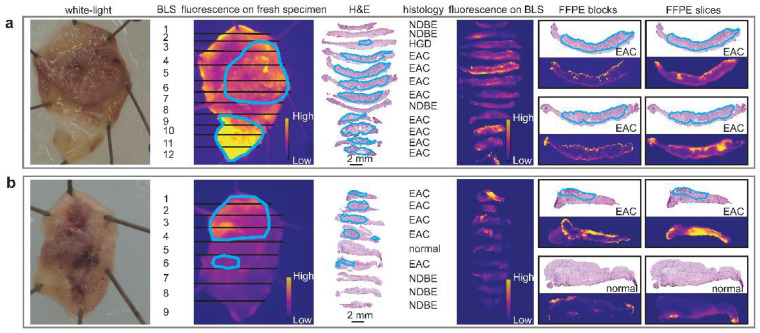
The ex vivo imaging and histology correlation of two novel tracers. The histology correlations of SULF1-800CW (**a**) and SPARC-800CW (**b**) on fresh EMR specimens, bread-loaf slices, FFPE blocks, and 4 µm FFPE slices are shown. The dysplastic or neoplastic BE tissue is delineated with blue circled lines. The fluorescence of both SULF1-800CW and SPARC-800CW tracers correlates well with the histology. BLS: bread-loaf slices; EAC: esophageal adenocarcinoma; FFPE: formalin-fixed and paraffin-embedded; H&E: hematoxylin and eosin; HGD: high-grade dysplasia; NDBE: non-dysplastic Barrett’s esophagus; normal: normal squamous epithelium.

**Figure 4 cancers-14-02462-f004:**
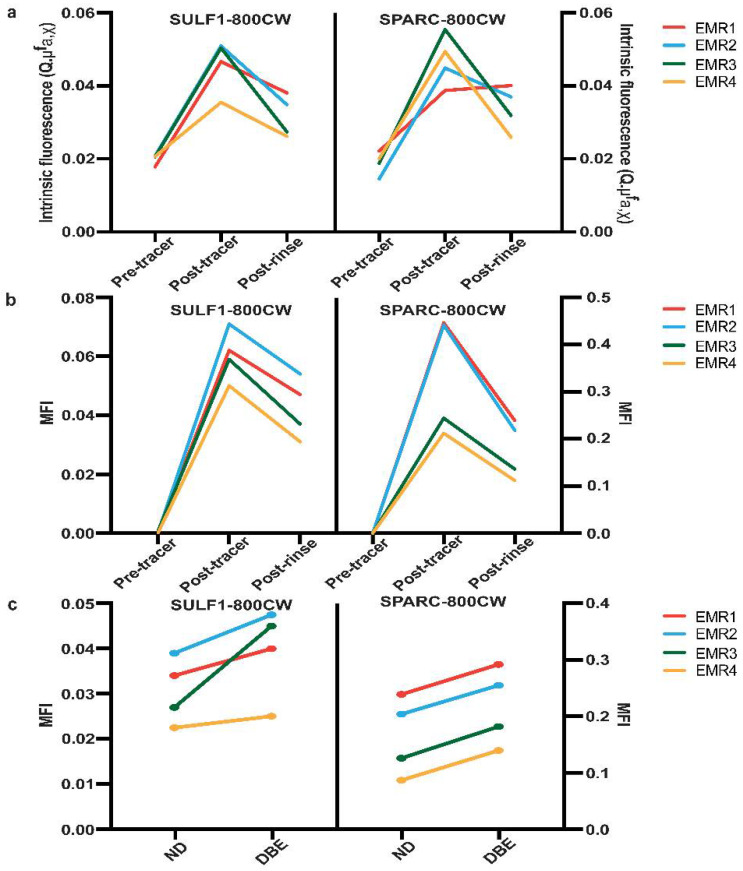
The fluorescence intensity analysis of SULF1-800CW and SPARC-800CW on EMR specimens. (**a**) The quantified intrinsic fluorescence with MDSFR/SFF spectroscopy of SULF1-800CW (left) and SPARC-800CW (right) before tracer spraying (pre-tracer), after tracer spraying, and before rinsing (post-tracer) and after rinsing (post-rinse). (**b**) The mean fluorescence intensity (MFI) of SULF1-800CW (left) and SPARC-800CW (right) before tracer spraying (pre-tracer), after tracer spraying before rinsing (post-tracer), after rinsing (post-rinse). (**c**) The MFI of dysplastic BE (DBE) tissue compared to non-dysplastic (ND) tissue. Each line connects paired data from the same specimen. DBE: dysplastic Barrett’s esophagus; EMR: endoscopic mucosal resection; MFI: mean fluorescence intensity; NDBE: non-dysplastic Barrett’s esophagus.

**Table 1 cancers-14-02462-t001:** Immunohistochemistry results of four targets in dysplastic Barrett’s esophagus.

Targets	H-Score DBE			High (201–300)		Intermediate (101–200)		Negative–Low (0–100)	
	*n*	Mean (SD)	Mean Value	*n*	%	*n*	%	*n*	%
SULF1	138	169 (41)	Intermediate	21	15.22	111	80.43	6	4.35
PKCι	137	129 (51)	Intermediate	15	10.95	70	51.09	52	37.96
DDR1	137	197 (42)	Intermediate	53	38.69	83	60.58	1	0.73
SPARC	138	175 (45)	Intermediate	38	27.54	93	67.39	7	5.07

H-score of 0–100 is classified as negative–low; H-score of 101–200 is classified as intermediate; H-score of 201–300 is classified as high. Mean intensities and standard deviation are shown using the H-score. DBE: dysplastic Barrett’s esophagus; SD: standard deviation.

**Table 2 cancers-14-02462-t002:** Immunohistochemistry results of four targets in non-dysplastic Barrett’s esophagus.

Targets	H-Score NDBE			High (201–300)		Intermediate (101–200)		Negative–Low (0–100)	
	*n*	Mean (SD)	Mean Value	*n*	%	*n*	%	*n*	%
SULF1	59	55 (33)	Negative–low	0	0	4	6.78	55	93.22
PKCι	56	67 (31)	Negative–low	0	0	8	14.29	48	85.71
DDR1	59	137 (50)	Intermediate	5	8.47	46	77.97	8	13.56
SPARC	58	120 (59)	Intermediate	4	6.90	29	50	25	43.10

H-score of 0–100 is classified as negative–low; H-score of 101–200 is classified as intermediate; H-score of 201–300 is classified as high. Mean intensities and standard deviation are shown using the H-score. NDBE: non-dysplastic Barrett’s esophagus; SD: standard deviation.

## Data Availability

All data generated or analyzed during this study are included in this published article or its Appendix A and available from the corresponding author on reasonable request.

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
