# Peer review of "Validation of Novel Molecular Imaging Targets Identified by Functional Genomic mRNA Profiling to Detect Dysplasia in Barrett’s Esophagus"

_cancers, 2022, doi:10.3390/cancers14102462_

Round 1

Reviewer 1 Report

The paper by Zhao et al is an ambitious attempt to put together many results of both literature search and practical experimental work, including development of novel labeled markers. All in all, the paper is clear, well written and interesting. However, the huge amount of different data makes the paper a bit hard to penetrate and get a clear overview of the scope and message. I suggest removing Table 1 and 2 as well as Figure 1 and Figure S2 and mention as data not shown.

I also would like to see a much more extensive part in the introduction describing the protein/mRNA markers and the reason for that work, as well as the aim with the labeling. Also, references are very few in that part.

Reviewer 2 Report

I thank the Editor for submitting this work to me for review.
The topic is interesting and useful but there are some inaccuracies in the definitions and in the structuring of the paper.

Introduction needs more precision in definitions. There is a bit of confusion in the description of materials and methods: sometimes Authors talk about genes, sometimes about mRNA and sometimes about proteins. It should be better described what is being studied and with what method (transcripts for mRNA, IHC for proteins, NGS for genes). The definitions of Barrett's esophagus, dysplasia and Barrett's adenocarcinoma are incorrect.
The pT classifications according to WHO and UICC 2017 are not correct.
The methodology of researching data in Literature is not clear.
The method of studying mucosectomies is unclear.

Finally, please review the annotations in the attached file.

Round 2

Reviewer 2 Report

The paper has been improved.